**Impact of aerosol-radiation interaction on new particle formation**

Gang Zhao[1], Yishu Zhu[1], Zhijun Wu[1], Taomou Zong[1], Jingchuan Chen[1], Tianyi Tan[1], Haichao Wang[1],

Xin Fang[1], Keding Lu[1], Chunsheng Zhao[2], Min Hu[1*]

1 State Key Joint Laboratory of Environmental Simulation and Pollution Control, International Joint

Laboratory for Regional Pollution Control, Ministry of Education, College of Environmental Sciences

and Engineering, Peking University, Beijing, 100871, China

Department of Atmospheric and Oceanic Sciences, School of Physics, Peking University, Beijing,

100871, China

*Correspondence author: Min Hu (minhu@pku.edu.cn)

**Abstract**

New particle formation (NPF) is thought to contribute to half of the global cloud condensation

nuclei. A better understanding of the NPF at different altitudes can help assess the impact of NPF on

cloud formation and corresponding physical properties. However, NPF is not sufficiently understood

in the upper boundary layer because previous studies mainly focus on ground-level measurements. In

this study, the developments of aerosol size distribution at different altitudes are characterized based

on the field measurement conducted in January 2019, in Beijing, China. We find that the partition of

nucleation mode particles at the upper boundary layer is larger than that at the ground, which implies

that the nucleation processing is more likely to happen in the upper boundary layer than that at the



ground. Results of the radiative transfer model show that the photolysis rates of the nitrogen dioxide

and ozone increase with altitude within the boundary layer, which lead to a higher concentration of

sulfuric acid at the upper boundary layer than that at the ground. Therefore, the nucleation processing

in the upper boundary layer should be stronger than that at the ground, which is consistent with our

measurement results. Our study emphasizes the influence of aerosol-radiation interaction on the NPF.

These results have the potential to improve our understanding of source of cloud condensation nuclei

in global scale due to the impacts of aerosol-radiation interaction.

## 1 Introduction

Atmospheric particles influence the earth's energy balance by directly interacting with the solar

radiation and indirectly being activated as cloud condensation nucleation (CCN) (Ghan and Schwartz,

2007). New particle formation (NPF) in the atmosphere and the herein coagulation may enable

particles to grow larger than 60 nm, at which point aerosols can exert radiative effects on the solar

radiation and act as CCN (Williamson et al., 2019;Shang et al., 2020). Some researchers find that the

NPF is responsible for around half of the global CCN (Merikanto et al., 2009;Du et al., 2017;Kulmala

et al., 2014). However, there is still considerable uncertainty about the magnitude that the NPF attribute

to CCN (Kulmala et al., 2004;Merikanto et al., 2009;Zhang et al., 2012). A better understanding of the

NPF at different altitudes can help assess the impact of NPF on cloud formation and corresponding

radiative effects. However, the underlying mechanism of NPF at different altitudes was not well

studied yet.





Nucleation requires sufficient amounts of precursor gases (Kulmala et al., 2004). Sulfuric acid

($H_2SO_4$) is thought to be the most important precursor for NPF events (Weber et al., 1997, 1996;Weber

et al., 2001;Stolzenburg et al., 2005;Kulmala and Markku, 2013). Knowledge in the profile of $H_2SO_4$

number concentrations ($[H_2SO_4]$) can help understand the NPF mechanism, while the profile of the

sulfuric acid is not well known due to the limitation of measurements.

The content of $H_2SO_4$ in a pseudo-steady state can be calculated (Kulmala et al., 2001) with:

$$[H_2SO_4] = k[OH][SO_2]/CS \tag{1}$$

Where $[OH]$ and $[SO_2]$ are the number concentrations of hydroxyl radical and sulfur dioxide,

respectively; CS is the condensation sink, which quantifies the limitation of NPF from existing

particles. It is calculated as (Maso et al., 2005):

$$CS = 2\pi D \sum \beta_m (D_{p,i}) D_{p,i} N_i \tag{2}$$

where $N_i$ is the particle concentration in the size $D_{p,i}$. The D is the diffusion coefficient of the $H_2SO_4$

and the $\beta_m$ is the transition regime correction factor. The $[OH]$ is related to solar ultraviolet radiation

(Rohrer and Berresheim, 2006). Previous studies found that the profile of photolysis radiation varies

significantly for different aerosol vertical distributions and the ultraviolet radiation is highly related to

the aerosol optical properties (Tao et al., 2014). Therefore, the ambient aerosol-radiation interaction

may exert a significant influence on the NPF through determining the [OH] vertical profile. However,

the influence of ultraviolet radiation on the NPF is not well understood.

In the past few decades, extensive measurements have been conducted at ground level to characterize the ambient aerosol particle number size distribution (PNSD) and then NPF events (Bullard et al.;Du et al., 2018;Peng et al., 2017;Malinina et al., 2017). Some studies suggest that the nucleation of fine particles can be altitude-dependent (Shang et al., 2018). High concentrations of

nucleation-mode particles were found in the upper parts of the boundary layer (Schobesberger et al., 2013). It is observed that the particle growth rate in the upper boundary layer is larger than that on the ground (Du et al., 2017). Measurements from the tethered balloon also show that a large partition of 11-16 nm particles was generated from the top region of the boundary layer, and was then rapidly mixed down throughout the boundary layer (Chen et al., 2018;Platis et al., 2016). Aircrafts

measurements (Wang et al., 2016;Zhao et al., 2020) also found that free troposphere favors the NPF. Most of these studies, to our best knowledge, focus on the concentration of precursor gases, but not on the aerosol-radiation interaction.

In this study, we first demonstrate that the NPF is more likely to happen in the upper boundary layer than in the near-ground surface layer based on field measurement of the aerosol PNSD profiles.

We find that the tendency of NPF is well related to ultraviolet radiation, implying that the aerosol-radiation interaction is an important factor that influences the NPF.

## 2 Data and Methods

### 2.1 Field Measurement



The field measurements were carried from 17 to 19 January 2019 at the Institute of Atmospheric

Physics (IAP), Chinese Academy of Sciences (39°18' N, 116°22' E), an urban site in Beijing China.

Details of the measurement site can refer to Wang et al. (2018). Vertical measurements were conducted

from the tower-based platform, with a maximum of 350 m, on the IAP campus. All of the instruments

were installed on a moving cabin of the tower, which moves up and down in altitudes between 0 and

240 m. The cabin moved around 10 meters every minute in altitude. Aerosol PNSD in the size range

between 10 nm and 700 nm were measured using a scanning mobility particle size (SMPS; TSI Inc.

3010). Aerosol scattering coefficient ($\sigma_{sca}$) at the wavelength of 450 nm, 525 nm, and 635 nm were

measured by an Aurora 3000 nephelometer (Müller et al., 2011). The nitrogen dioxide (NO2) was

measured based on its absorbance at 405 nm with a low-power lightweight instrument (model 405 nm,

2B Technology, USA). The nitrogen monoxide (NO) was measured by adding an excess of ozone with

another power lightweight instrument (model 106-L, 2B Technology, USA). The wind speed, wind

direction, ambient relative humidity, and temperature were measured by a small auto meteorology

station. This instrument can record the atmosphere pressure, which was used to retrieve the altitude

information.

**2.2 Lognormal fit of PSND**

For each of the measured PNSD, it is fitted by three lognormal distribution modes by:

$$N(Dp) = \sum_{i=1,2,3} \frac{N_i}{\sqrt{2\pi}\log(\sigma_{g,i})} exp\left[-\frac{\log(Dp)-\log(Dp_i)}{2log^2(\sigma_{g,i})}\right] \tag{3}$$



Where $N_i$, $\sigma_{g,i}$, and $Dp_i$ are the number concentration, geometric standard deviation, and geometric mean diameter of mode $i$ respectively. Two examples of fitting the measured PNSD are shown in Fig. S1. The three modes with geometric diameter ranges of 10 ~ 25 nm, 25 ~100 nm, and 100 ~ 700 nm

correspond to the nucleation mode, Aitken mode, and accumulation mode respectively. The nucleation particles mainly result from the nucleation process and the Aitken mode particles are from primary sources, such as traffic sources (Shang et al., 2018). The accumulation mode particles are correlated with secondary formation, which mainly represents the ambient pollution conditions (Wu et al., 2008).

## 2.3 Mie Model

Mie scattering model (Bohren and Huffman, 2007) is used to estimate the aerosol optical properties. When running the Mie model, aerosol PNSD, aerosol black carbon mass size distribution and refractive index are essential. The measured mean black carbon mass size distribution from Zhao et al. (2019) is adopted in this study, which is measured around 3 kilometers away from this site. The refractive index of the non-black carbon and black carbon aerosol component are 1.64+0i, which is

the measured mean aerosol refractive index measured at Beijing (paper in preparation), and 1.96 + 0.66i (Zhao et al., 2017) respectively. The aerosol hygroscopic growth is not considered here because the ambient relative humidity during the measurement was all the way lower than 30% as shown in fig. 1(b). With the measured different aerosol PNSD and above-mentioned information, we can calculate the corresponding aerosol optical properties, which contain the aerosol $\sigma_{sca}$, aerosol single

scattering albedo (SSA) and asymmetry factor (g).

### 2.4 TUV Model

The Tropospheric Ultraviolet-Visible radiation model (TUV), developed by Madronich and Flocke (1997), is an advanced transfer model with an eight-stream, discrete ordinate solver. This model can calculate the spectral irradiance, spectral actinic flux, and photo-dissociation frequencies in the wavelength range between 121 nm and 735 nm. In this study, the photolysis frequency of the nitrogen dioxide ($J(NO_2)$) and ozone ($J(O^1D)$) were used for further study. Inputs of the TUV model are the aerosol optical depth and single scattering albedo (Tao et al., 2014). The cloud aerosol optical depth is set to be zero in this study. The output of the TUV model includes the profiles of $J(NO_2)$ and $J(O^1D)$.

Some changes were made in the source code of the TUV model so that the model can calculate the $J(NO_2)$ and $J(O^1D)$ profiles with different aerosol optical profiles (including aerosol $\sigma_{sca}$, SSA, and g).

### 3 Results and Discussions

### 3.1 Aerosol PNSD at different altitude and time

The measured aerosol PNSD profiles in the time range between 7:00 and 18:50 on January 18 were used for analysis, which contained eight different upward movement and downward movement of the cabin, respectively. Fig. 1 (a) gives detailed time-altitude information of each measurement. All of the time mentioned in the research corresponds to the local time zone.

On January 18, the measured ambient temperature and relative humidity ranges were -3ºC ~ 10ºC

and 13% ~ 24% respectively, which implied that the ambient air in the winter of Beijing are dry and

cold. Aerosol hygroscopic growth was thus not considered in this study. The wind speeds during the

measurement were lower than 1m/s, and thus the measurement results of aerosol microphysical

properties were hardly influenced by transportation.

During the measurement, the $\sigma_{sca}$ varied between 0 and 400 Mm$^{-1}$. It ranged between 100 Mm$^{-1}$

and 200 Mm$^{-1}$ on 18, January. We compared the measured $\sigma_{sca}$ using the nephelometer and calculated

$\sigma_{sca}$ using the Mie scattering model and measured PNSD. The measured and calculated $\sigma_{sca}$ show

good consistency with slope values of 1.00, 0.95, and 0.89 for wavelengths of 450 nm, 525 nm, and

635 nm, respectively, as shown in Fig. S2. The calculated $\sigma_{sca}$ values are slightly smaller than that

of the measured ones because the measured aerosol PNSD only covers the aerosol diameter between

10 nm and 700 nm, while the measured $\sigma_{sca}$ represents the optical properties of the whole population.

The square of the correlation coefficients are 0.97, 0.97, and 0.97 for the above-mentioned different

wavelengths. Our results demonstrate that the measured ambient aerosol PNSDs are reliable for further

analysis.

The measured aerosol PNSD varied significantly for different altitudes and a different time. PNSD

profiles in Fig. 2 corresponded to these time periods when the cabin moved upward. The corresponding

downward PNSD profiles are shown in Fig. S2. In the early morning, the PNSD on the ground surface

is substantially different for different altitudes. Particle number concentration on the ground surface

can reach $1.5 \times 10^4$ cm$^{-3}$, and the number concentrations peaked at smaller than 100 nm. It was only $8 \times 10^3$ cm$^{-3}$ with peaking aerosol diameter at around 200 nm at a higher altitude around 200 m. The solar radiation in the morning was very week, therefore, the turbulence mixing of the aerosol among

different altitudes was very weak. The initial emission from the ground surface cannot be mixed up to higher locations, and thus the aerosol number concentrations at the surface was larger than that at a higher level as shown in Fig. 2(a).

With the increment of solar radiation and ambient temperature, the turbulence mixing of ambient particle became stronger. The aerosol PNSD at the surface decreased with time because the near

ground particles were mixed up to a higher location as shown in Fig. 2(b) and (c). However, the aerosol PNSD at higher altitude increased with time due to the upcoming mixed aerosol particles from lower altitude. Therefore, the difference between the aerosol PNSD at different altitudes became smaller with the development of the boundary layer as shown in Fig. 2 (b), (c), and (d). These particles were still not well mixed at the range between 0 and 240 m until 11:20.

In the afternoon, the boundary layer was well mixed with the increment of solar radiation and ambient temperature. The aerosol PNSD and PVSD were almost uniformly distributed as shown in Fig. 2 (e) and (f). However, the turbulence was relatively weak after 15:00 as the measured PNSD and PVSD on the ground surface were slightly larger than that of a higher place. After 16:00, the turbulence was weaker because a larger difference between the PNSD at the ground surface and the higher level

existed. The ambient particles were hardly mixed after the sunset. The measured aerosol PNSD profiles

showed almost the same properties as that in the morning, with more aerosol particles located on the ground surface from emissions.

Overall, the measured PNSD profiles were highly related to the intensity of turbulence. When the turbulence was weak, the PNSD at surface was different from that of upper levels because the initially

emitted particles cannot be mixed up to higher location. The PNSD tended to be uniformly distributed when the turbulence within the boundary layer was strong.

**3.2 Nucleation process in the upper boundary layer**

We calculated aerosol total number concentration for each measured PNSD ($N_{tot}$) and the profiles of $N_{tot}$ are shown in Fig. 3 (a). All of the profiles in Fig. 3 corresponded to these cases when the cabin

is moving up. The $N_{tot}$ profiles varied significantly with the development of the boundary layer. In the morning, the $N_{tot}$ in the surface (larger than $2\times10^4$ cm$^{-3}$) was larger than that at a higher level (lower than $1\times10^4$ cm$^{-3}$) because the turbulence is so weak that the initially emitted particles on the surface cannot be transported to the upper level. In the afternoon around 14:00 and 16:00, the aerosol was well mixed in the boundary layer and $N_{tot}$ was almost uniform with around $1.2\times10^4$ per cubic centimeter.

Afterward, the turbulence was weaker than that in the early afternoon and again the emitted aerosols cannot reach the higher level. The profile of $N_{tot}$ in the morning was similar to that in the late afternoon and night.

The number ratio profiles of nucleation mode to Aitken mode ($N_1/N_2$) for different times are shown in Fig. 3(b) and summarized in Table 1. In the morning of 7:00, the ratio decreased from around

0.6 to 0.04 when the cabin moved up from 0 to 240 m. The ratio on the ground surface decreased

because the temperature and turbulence increased when it came to 8:00-10:00 in the morning. However,

the turbulence was not strong enough to mix all of the particles to upper levels to 240 nm. The ratio

still decreased with altitude. In the afternoon, the boundary layer developed well and the ratios between

13:20 and 14:25 were almost uniformly distributed at different altitudes. However, we found that the

ratio increased with altitude from 0.21 to 0.34 when it came to 16:15, which implied that more

nucleation mode particles were formed in the upper level in the boundary layer. The increment of the

ratio was hardly influenced by transportation because the wind speed during the measurement was all

the time lower than 1 m/s as shown in Fig. 1(b).

To better configure the variations of PNSD, we calculated the aerosol number concentrations with

the diameter between 10 and 25 nm ($N_{10-25nm}$). The $N_{10-25nm}$ profiles in Fig. 3(c) show almost the same

trends with the number ratio of $N_1$ to $N_2$. In the morning and later afternoon, the $N_{10-25nm}$ decreased

with the altitude. The $N_{10-25nm}$ in the early afternoon were uniformly distributed due to the strong

mixing in the boundary layer. When it came to 16:15, the $N_{10-25nm}$ at different altitudes were larger

than that in the early afternoon. Most importantly, $N_{10-25nm}$ increases with altitude. The aerosol total

volume at 16:15 does not increase with altitude because the nucleation produced particles are so small

that they contribute negligibly to the aerosol total volume.

Based on the discussion above, we found that the total aerosol number concentrations increased

slightly with altitude at 16:15. The number ratio of $N_1$ to $N_2$ and the $N_{10-25nm}$ increased with altitude.

The total volumes of the aerosol particles were almost the same at different altitudes. The variation of

PNSD was hardly influenced by transportation. Therefore, we concluded that the nucleation process

was more likely to happen in the upper level of the boundary layer than the ground surface. This

phenomenon was not observed in the early afternoon because the turbulence in the early afternoon is

so strong that the aerosol particles are well mixed in the boundary layer.

Many previous studies have reported the NPF events in the upper boundary layer. The study in

Platis et al. (2016) reported that the NPF originated at elevated altitude, and then being mixed down to

the ground in Germany. The higher nucleation mode particle number concentrations were observed at

the top region of the boundary layer and were then rapidly mixed throughout the boundary layer in

America (Chen et al., 2018). Qi et al. (2019) also found the NPF at the top of the boundary layer based

on tethered airship measurements in eastern China. The NPF events were also observed at different

altitude in the North China Plain (Zhu et al., 2019).

**3.3 Influence of Aerosol-radiation Interaction on NPF**

Based on equation 1, the nucleation rate mainly depends on [OH], [$SO_2$], and CS. The [$SO_2$] is

not available at this measurement. However, we measured the [$NO_x$], which is the sum of NO and $NO_2$.

The profiles of the [$SO_2$] and [$NO_x$] should be the same because both of them are mainly emitted from

the ground and then mixed up by turbulence. The [$NO_x$] in the afternoon is almost uniformly

distributed as shown in Fig. 4(a). Thus, the [$SO_2$] should be uniformly distributed in the afternoon

within the boundary layer. The CS profiles, in Fig. 4(b), were almost uniformly distributed in the

afternoon. Therefore, the [OH] is the only main factor that may result in different characteristics of



NPF at different altitudes. Based on the work of Ehhalt and Rohrer (2000), the [OH] can be calculated

by:

$$[OH] = a[J(O^1D)]^\alpha [J(NO_2)]^\beta \frac{b[NO_2]+1}{c[NO_2]^2+d[NO_2]+1}$$ (4)

With $\alpha$, $\beta$, $a$, b, c, d equaling 0.83, 0.19, 4.1x10$^9$, 140, 0.41, and 1.7, respectively. From equation 4,

the vertical distribution of $J(O^1D)$, and $J(NO_2)$ played a significant influence on [OH] and further

influence the NPF. However, the $J(O^1D)$, and $J(NO_2)$ were not measured. The TUV model was

employed to estimate the $J(O^1D)$, and $J(NO_2)$ for different aerosol profiles.

The input of the TUV needs the aerosol optical properties in the altitude range between 0-20 km.

The parameterization of aerosol number concentration profiles by Liu et al. (2009) with aircraft

measurement in Beijing is used in this study. Liu et al. (2009) found that number concentration constant

within the boundary layer, linearly decreasing within the transition layer and exponential decreasing

above the transition layer, when the particles within the boundary are well mixed. The normalized

aerosol PNSD (PNSD divided by total aerosol number concentration) was assumed to be the same at

different altitudes. The BC to total aerosol mass concentration ratio was also assumed to be the same

at different altitudes (Ferrero et al., 2011). The $\sigma_{sca}$, SSA, and g profiles can be calculated by Mie

theory under these assumptions (Zhao et al., 2017;Zhao et al., 2018).

The lines with squares in Fig. 5(a) and (b) provide the calculated photolysis rates of $J(O^1D)$, and

$J(NO_2)$ with a boundary layer altitude of 1000 m. Results show that both the $J(O^1D)$, and $J(NO_2)$

increase with altitude within the boundary layer. The $J(O^1D)$ increases from 8.9x10$^{-3}$ s$^{-1}$ to 14.3x10$^{-}$

$^3$ s$^{-1}$ and $J(NO_2)$ increases from 3.0x10$^{-5}$ s$^{-1}$ to 6.2x10$^{-5}$ s$^{-1}$ in the boundary layer. The corresponding

[OH] increased from 6.2x10$^6$ cm$^{-3}$ to 11.9x10$^6$ cm$^{-3}$ based equation 4. Thus, the [OH] at the top of the

boundary layer is two times of that on the ground surface due to the variation in photolysis rate. Our

estimated [OH] at the surface is consistent with the previously estimated relationships between the

[OH] and $J(O^1D)$ (Rohrer and Berresheim, 2006).

Overall, the aerosol profiles tend to be uniformly distributed within the boundary layer due to the

strong turbulence in the afternoon. The corresponding estimated $J(O^1D)$, and $J(NO_2)$ values

increase with altitude, which leads to higher [OH] at the top of the boundary layer than that at the

ground. Therefore, the [H$_2$SO$_4$] should increase with altitude based on equation 1. There should be

more nucleation processing at the top of the boundary layer than that at the ground, which is consistent

with our field measurement. The schematic graph of the influence of aerosol-radiation interaction on

NPF is shown in Fig. 6.

**3.4 Impact of Boundary layer development on the photolysis rates**

For a better understanding of the aerosol-radiation interaction on NPF, we estimated the photolysis

rates under different aerosol vertical profiles. Based on the work of Liu et al. (2009), two typical types

of aerosol profiles exists under different boundary layer as shown in Fig. S4. For the first type of

boundary layer, aerosols were not well mixed within the boundary layer and the aerosol number

concentrations decrease with altitude exponentially (type A). Another type of boundary layer has

aerosol number concentration constant in the boundary layer and then decreasing with altitude above



the boundary (type B). For type B, we estimated the corresponding photolysis rate for different boundary layer heights between 500 m and 1000 m, which covers the mean boundary layer altitude in the North China Plain (Zhu et al., 2018). The different aerosol optical depth (AOD), which ranges

between 0.3 and 2, are used for different pollution conditions.

Four different aerosol profiles are used in this study. Details of the four different aerosol profiles are summarized in Table 2. The first one corresponds to the aerosol boundary layer type A, with a boundary altitude of 1000 m and AOD of 0.3 (B1). The second aerosol profile has the same boundary altitude of 1000m and AOD of 0.3, but the boundary layer type is changed into B (B2). The third

aerosol profile also corresponds to boundary layer type B, and a boundary layer altitude of 1000m, but the AOD is 0.8 (B3). The last one has a boundary layer altitude of 500m, with an AOD of 0.8 and a boundary layer type of B (B4).

The $J(O^1D)$, and $J(NO_2)$ profiles under the above-mentioned aerosol profiles are estimated and shown in Fig. 5 (a) and (b). For each type, both the $J(O^1D)$, and $J(NO_2)$ increase with altitude. The

increased ratio of the $J(O^1D)$ with altitude ($k_{O^1D}$) are $1.7 \times 10^{-5}$, $2.0 \times 10^{-5}$, $3.0 \times 10^{-5}$, and $5.4 \times 10^{-5}$ $s^{-1}km^{-1}$, for the aerosol profile of B1, B2, B3, and B4 respectively. The corresponding increase ratio of the $J(NO_2)$ with altitude ($k_{NO_2}$) are $2.6 \times 10^{-3}$, $3.3 \times 10^{-3}$, $5.3 \times 10^{-3}$, and $9.0 \times 10^{-3}$ $s^{-1}km^{-1}$, for B1, B2, B3, and B4, respectively. The increase ratio of [OH] were estimated to be $3.4 \times 10^6$, $4.1 \times 10^6$, $5.5 \times 10^6$, and $7.4 \times 10^6$ $cm^{-3}km^{-1}$ for B1, B2, B3 and B4, respectively (Table 2).



These four profiles represent the typical ambient aerosol profiles in the early morning, late

morning, early afternoon, and late afternoon, respectively. In the early morning, the turbulence in the

boundary layer is weak and the aerosol within the boundary layer is not well mixed (B1). In the late

morning, the aerosol in the boundary is well mixed and uniformly distributed due to the increasing

turbulence (B2). The early afternoon (B3) should have higher AOD when compared with that in the

late morning due to the formation of the secondary aerosol. However, the boundary layer altitude

decreased in the late afternoon (B4) because the turbulence within the boundary layer weakened

compared with B3. The ambient aerosol profiles tend to change from B1 to B4 from early morning to

late afternoon. The corresponding $k_{O^1D}$ and $k_{NO_2}$ increased with the development of the boundary

layer. In the late afternoon, the difference of photolysis rate at the top of the boundary layer and ground

are largest. Furthermore, the turbulence in the mixing layer is weakened and the nucleation formed

particles cannot be mixed down to the ground. Therefore, it is more likely to observe more nucleation

mode particles at the top of the boundary layer than at the ground in the late afternoon, which is

consistent with our measurement.

## 4. Conclusion

In this study, we characterized the aerosol PNSD at different times and different altitudes based

on field measurements at a urban site, in Beijing, China. Our measurements show that the aerosol size

distribution profiles varied significantly with the development of the boundary layer.

In the morning, the turbulence in the boundary was weak and the initial emitted particles cannot

be mixed to a higher layer. The corresponding aerosol PNSD at the surface was larger than that at

higher locations. At noon, the particles within the boundary were well mixed and tend to be uniformly

distributed at different altitudes. In the late afternoon, we found more nucleation mode particles at a

higher altitude than that at the ground. The larger partitions of nucleation mode particles do not result

from transformation. We concluded that the nucleation processing in the upper boundary layer were

more likely to happen than that at the ground.

The TUV model was employed to estimate the profile of photolysis rate for different aerosol

profiles. Results showed that both the $J(O^1D)$, and $J(NO_2)$ values increased with altitude, which led

to higher [OH] at the top of the boundary layer than that at the ground. The corresponding [H₂SO₄]

should increase with altitude based on equation 1, when the aerosol was well mixed and uniformed in

the mixed layer. Therefore, more nucleation processing at the top of the boundary layer may happen

than that at the ground, which is consistent with our field measurement.

We also estimate the corresponding photolysis rate profile under different boundary structures.

The increasing ratio of the photolysis rate with altitude increase with the development of the boundary

layer from early morning to late afternoon. In the late afternoon, the difference of the photolysis rate

at the upper boundary layer and that at the ground are the largest. At the same time, the turbulence is

not so strong that the nucleation mode particles formed in the upper boundary layer are not able to mix

down to the ground. Therefore, it is a favor to observe higher nucleation mode particles concentration

at the upper boundary layer than that at the ground in the afternoon. Our study reveals that the vertical



distribution of ambient aerosols would first influence the vertical profile of the photolysis rate. Then

the NPF for different altitudes is tuned due to the different photolysis rates.


*Data availability.* The data is available upon request to the corresponding author.

*Author contributions.* Gang Zhao and Yishu Zhu did the analysis and wrote the manuscript. Min Hu, Chunsheng Zhao, Zhijun Wu, Xin Fang, and Gang Zhao discussed the results. Yishu Zhu, Jingchuan Chen, Taomou Zong, Tianyi Tan, Keding Lu, and Haichao Wang conducted the measurements.

*Competing interests.* The authors declare that they have no conflict of interest.

*Acknowledgments.* This work is supported by the National Natural Science Foundation of China (91844301) and the National Key Research and Development Program of China (2016YFC0202000 Task 3, 5).




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



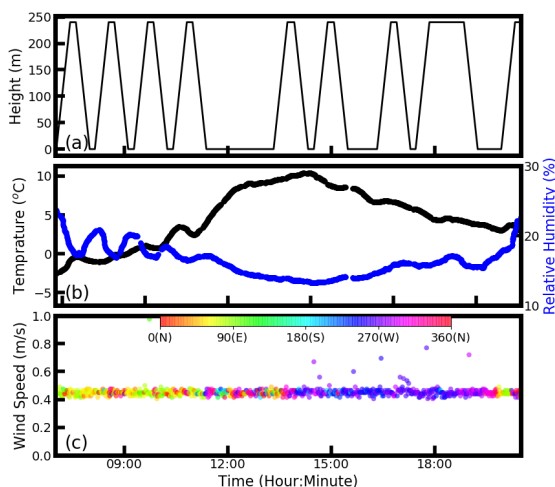


**Figure 1.** Time series of (a) the measurement altitude, (b) temperature (black line) and relative

humidity (blue line), and (c) the wind speed and wind direction.

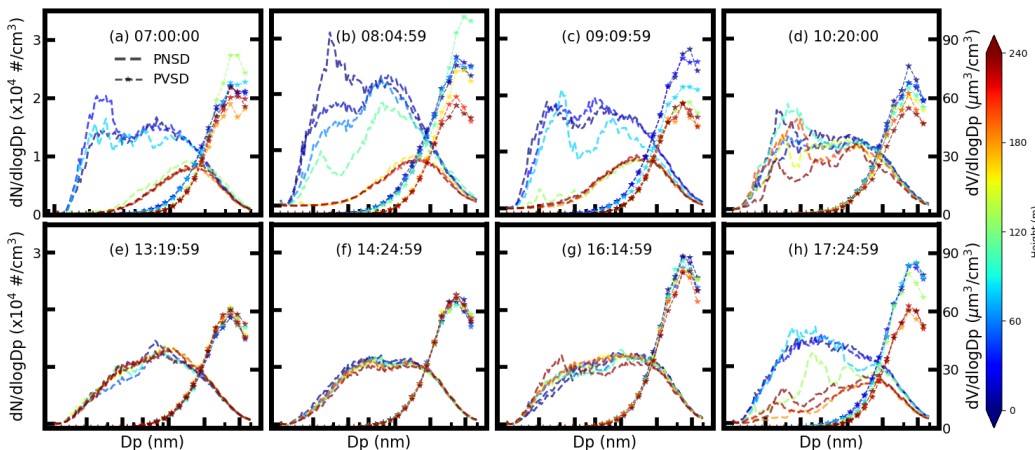

**Figure. 2.** The measured aerosol PSND (dashed line) and the PVSD (dashed line with star) at (a) 7:00,

(b) 8:05, (c) 9:50, (d) 10:20, (e) 13:20, (f) 14:25, (g) 16:15, and (h) 17:25. The filled colors represent

the corresponding measurement altitude above the ground.



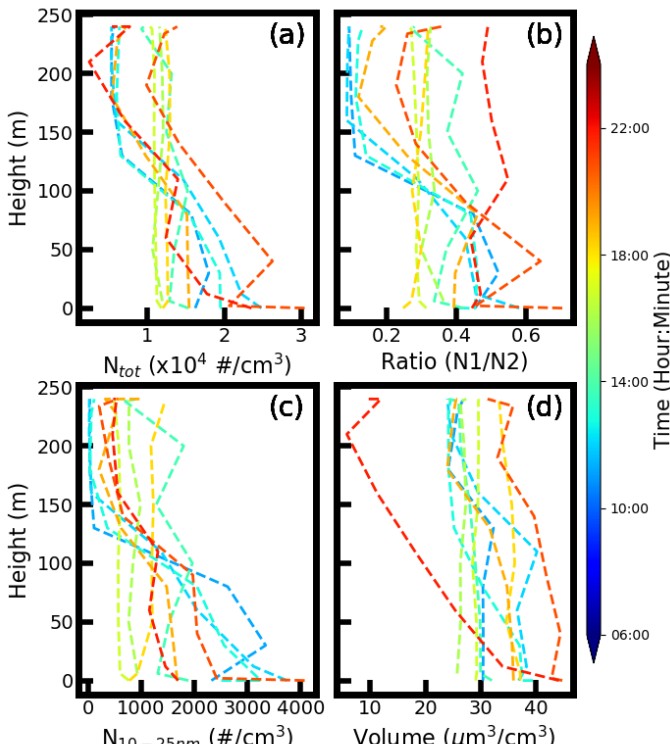

**Figure. 3.** The measured (a) aerosol number concentrations, (b) number ratio of the nucleation mode aerosol number concentrations to Aitken mode aerosol number concentrations, (c) aerosol number concentrations for 10-25 nm, and (d) measured aerosol volume concentrations at different altitudes. The filled colors of different lines denote the different measurement times.






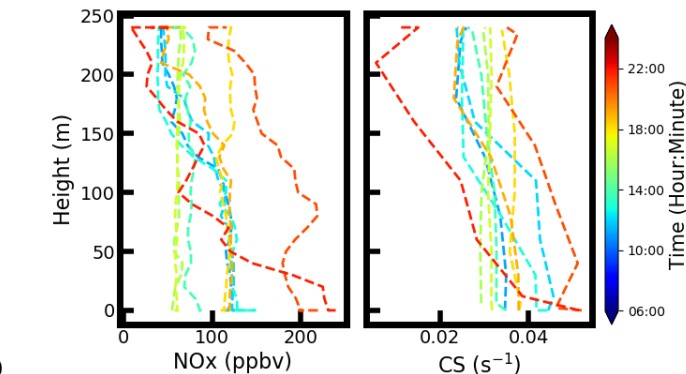

**Figure. 4.** The measured (a) $NO_x$ and (b) CS at different altitudes. The filled colors of different lines

denote the different measurement times.



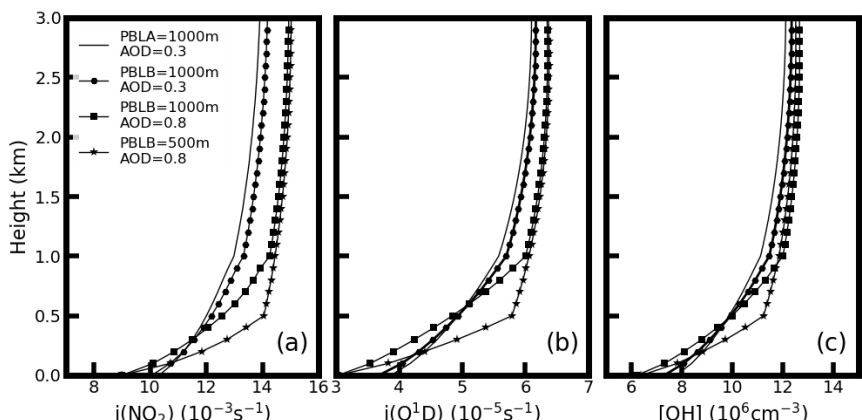

**Figure. 5.** The estimated (a) $j(NO_2)$, (b) $j(O^1D)$, and (c) OH concentration for different aerosol profiles.

The (1) solid line, (2) solid line marked with hexagon, (3) solid line marked with squares, and (4) solid

line marked with stars represent the aerosol distribution of B1, B2, B3, and B4, respectively.



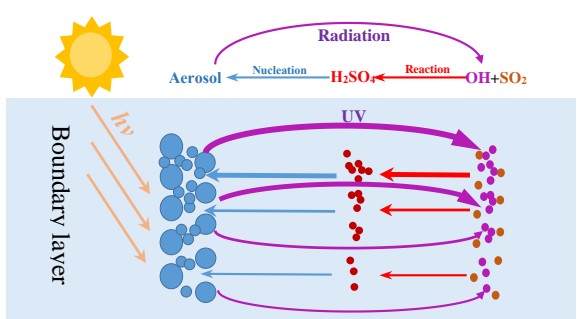

**Figure 6.** Schematic of the influence of aerosol-radiation interaction on NPF.






**Table 1.** The number ratio of nucleation mode to Aiken mode.

| Time<br>Altitude | 7:00 | 8:05 | 9:10 | 10:20 | 13:20 | 14:25 | 16:15 | 17:25 |
|---|---|---|---|---|---|---|---|---|
| 0 | 0.56 | 0.52 | 0.43 | 0.36 | 0.21 | 0.20 | 0.21 | 0.32 |
| 30 | 0.63 | 0.44 | 0.42 | 0.35 | 0.21 | 0.19 | 0.23 | 0.30 |
| 60 | 0.61 | 0.34 | 0.40 | 0.40 | 0.22 | 0.19 | 0.27 | 0.28 |
| 110 | 0.05 | 0.26 | 0.25 | 0.46 | 0.27 | 0.19 | 0.28 | 0.14 |
| 160 | 0.04 | 0.03 | 0.07 | 0.39 | 0.20 | 0.17 | 0.27 | 0.17 |
| 210 | 0.03 | 0.03 | 0.08 | 0.51 | 0.20 | 0.17 | 0.30 | 0.31 |
| 240 | 0.04 | 0.03 | 0.09 | 0.26 | 0.21 | 0.16 | 0.34 | 0.37 |






**Table 2.** Details of the aerosol optical profiles and estimated photolysis values.

| Profile | Type[*1] | Altitude[*2] | AOD | k [J(NO$_2$)] (10$^{-3}$ s$^{-1}$km$^{-1}$) | k [J(O$^1$D)] (10$^{-5}$ s$^{-1}$km$^{-1}$) | k [OH] (10$^6$ cm$^{-3}$km$^{-1}$) |
|---------|----------|--------------|-----|---------|---------|---------|
| B1 | A | 1000 | 0.3 | 2.6 | 1.7 | 3.4 |
| B2 | B | 1000 | 0.3 | 3.3 | 2.0 | 4.1 |
| B3 | B | 1000 | 0.8 | 5.3 | 3.0 | 5.5 |
| B4 | B | 500 | 0.8 | 9.0 | 5.4 | 7.4 |

[*1]**Boundary layer Type.**

[*2]**Boundary layer altitude.**