# Peer review of "Impact of aerosol-radiation interaction on new particle formation"

_Atmospheric Chemistry and Physics, 2020_

## Author Response (AR1)

Response to reviewer#1

Thanks for the reviewer's helpful and insightful suggestions! The comments are addressed point-by-point and responses are listed below.

**Comments:** New particle formation contributes to more than half of CCN and thereby is important for climate. This work conducts the vertical measurements of particle number size distribution on a moving cabin of a 350 m tower. This kind of experiment is interesting and can improve the understanding of the vertical distributions of new particle formation (NPF) in the urban environment. However, I believe that the evidence on the occurrence of nucleation at 240 m or upper boundary layer in this manuscript is not enough. A major revision is needed before considering publication in ACP.

**Reply:** Thanks for the comments.

**Comments:** Line 44-50: It is not suitable to put equations of sulfuric acid proxy and CS and the explanation of these two equations in the Introduction. I suggest moving those to Section 2.

**Reply:** Thanks for the comments and helpful suggestions. The corresponding equations were moved to section 2.5. We added a section to describe how the sulfate acid concentration can be influenced by the photolysis ratio in section 2.5.

**Comments:** Line 79-89: The time resolutions of instruments need to be presented. How do they compare with the lifting speed of the cabin?

**Reply:** Thanks for the comment. The time resolutions were added in the corresponding text. The cabin moved around 10 meters every minute in altitude. Aerosol PNSD was measured using a scanning mobility particle size every five minutes. Aerosol scattering coefficient ($\sigma_{sca}$) were measured by an Aurora 3000 nephelometer with a time resolution of one minute. The nitrogen dioxide (NO$_2$) was measured every minute based on its absorbance at 405 nm with a low-power lightweight instrument (model 405 nm, 2B Technology, USA). The nitrogen monoxide (NO) was measured by adding an excess of ozone with another power lightweight instrument (model 106-L, 2B Technology, USA) with a time resolution of one minute. All of the data were averaged with a time resolution of five minutes.

**Comments:** Line119-121: I suggest the author gives the description of the changes of the TUV model in SI.

**Reply:** Thanks for the comment. We added a brief description of the changes of the TUV model in the manuscript.

In the TUV model, the input of the aerosol optical properties are the aerosol optical depths at the wavelength of 550 nm and the column-averaged SSA. The profiles of the $\sigma_{sca}$ are calculated assuming that the aerosol $\sigma_{sca}$ are proportional to those measured by Elterman et al. (1968). The g values are set to be fixed as 0.61. Some changes were made in the source code of the TUV model. In our model, the author-defined aerosol $\sigma_{sca}$ profiles, SSA profiles and g profiles can be used as the input of the model. Therefore, the J(NO$_2$) and J(O$^1$D) profiles with different aerosol optical profiles (including aerosol $\sigma_{sca}$, SSA, and g) can be estimated.

**Comments:** Line 144-145: What are the differences in PNSD when cabin moving upward and downward? Because the time is close when cabin moving upward and downward, I would suppose the PNSDs are similar. If so, I suggest the author merges the upward and downward PNSD. If not, please explain the reasons.

**Reply:** Thanks for the comment. The PNSD when the cabin moving upward and downward is the time corresponds to different measurement time and the time interval differs for different height. The time shown in Fig. 2 (Fig. S3) corresponds to the time when the cabin begins to move up (down) from the ground (240 m height). The time intervals are approximately one hour and half an hour for the measured aerosol PNSD at the ground and the height of about 120 m respectively when the cabin moves upward and downward in one cycle. The aerosol PNSD may vary significantly within an hour. Therefore, it is not appropriate to merge the upward and downward PNSD. The main conclusions in the development of aerosol PNSD with time are almost the same when the cabin moves upward and downward, and thus we placed the measured PNSD when cabin moving upward downward in different figure.

**Comments:** Section 3.2: In this section, the authors discussed the stronger nucleation in the upper boundary layer. However, some more evidences are needed for this conclusion.

1. The maximum altitude of this vertical measurement is 240 m. The boundary layer height is around 1500 m in winter. Therefore, I don't think this measurement can represent the

situations of upper boundary layer even in winter. I suggest the author uses 'above the urban canopy' instead of the 'upper boundary layer'.

**Reply:** Thanks for the comment. We agree with the reviewer that it is not appropriate to use the 'upper boundary layer'. We replaced the "upper boundary layer" as the "upper mixing layer" in the manuscript for two reasons. The first is that the statistical results of the mixed layer height in Beijing winter is $493\pm131$ m (Zhu et al., 2018). The corresponding height is even lower during the haze episodes (Wang et al., 2018). Our measurement covers about half of the mixed layer height. The second is that despite that the maximum altitude of our vertical measurement is 240 m, our general framework of the development of the atmospheric mixing layer is consistent with our measurements and previous studies (Zhu et al., 2018). As long as the aerosol was uniformly distributed in the mixed layer due to the strong turbulence, our main conclusion that the nucleation processing in the upper mixed layer is stronger than that at the ground is applicable.

**Comments:** 2. I suggest the author give the legend of each profile in Fig.3.

**Reply:** Thanks for the helpful suggestions. We added the legend of each profile in Fig. 3 and Fig. 4.

**Comments:** 3. Are there any ground-based measurements on this day? From the ground-based measurements, is it a new particle formation event day?

**Reply:** Thanks for the comment. During the field campaign, there was no ground-based measurement of the aerosol PNSD. From the measured PNSD at the ground in Fig. 2, we think it is not a new particle formation event day.

**Comments:** 4. The ratio of nucleation mode particles number concentrations to Aitken mode particles number concentrations increased at 16:15. Does the author mean the nucleation occur at late afternoon? Most of NPF events start at noontime when the solar radiation is strong. Although the ratio increased at 16:15, the PNSD shown in Fig. 2 is not a typical PNSD of nucleation.

**Reply:** Thanks for the comment. We don't mean that the nucleation occurs in the afternoon. We agree with the reviewer that most of the NPF events start at noontime or in the morning when the solar radiation is strong. In fig. 2, we want to show that the measured ratio of the nucleation mode particle number concentrations to Aitken mode aerosol number concentration increase with height in the afternoon. The nucleation process in the upper boundary layer is stronger than that at the ground. The difference in the aerosol number concentration ratio of nucleation mode to Aitken mode is not observed in the noontime because the turbulence in the noontime is so strong that the particles in the vertical distribution are well mixed. The turbulence is rather weak in the late afternoon and then we can observe more aerosol number concentrations in nucleation mode in the late afternoon.

The fig. 2 is not a typical PNSD of nucleation because the measured PNSD results from the long differential mobility analyzer (DMA) at the size range smaller than 15 nm, where the PNSD is always underestimated. Our measurement results are in accordance with the

previous measurement of PNSD (Du et al., 2018;Qi et al., 2019). In the related reference, the PNSD is not a typical PNSD.

**Comments:** 5. Although the wind speed is low during the measurement, the wind direction changed at around 16:00. Is it possible that the change of the air masses caused the observed phenomenon? Are there intensive local anthropogenic emissions to the southwest of measurement site?

**Reply:** Thanks for the comments. The wind direction changed at around 14:00. The measured aerosol PNSDs were almost the same at 13:20, 13:50, 14:25, and 15:05 as the turbulence in the noon was strong and the aerosols were uniformly distributed. Thus, the observed phenomenon is not likely to be caused by the change of the air mass. The local anthropogenic emissions may influence the PNSD at the ground in the morning as shown in Fig. 2(a), (b). However, it is not likely that the local anthropogenic emissions may influence the observed phenomenon in fig. 2(e), (f) and (g).

**Comments:** Line219-220: The author needs to give some evidences or cite some references here. $SO_2$ can be from the power plant and the NOx is most from the vehicle emissions. They may have different vertical distributions.

**Reply:** Thanks for the helpful suggestions. We added some discussion in the corresponding manuscript. Both the NOx and $SO_2$ were mainly from the ground emission. The $SO_2$ tends to have a longer lifetime than that of NOx (Steinfeld, 1998). Thus, the $SO_2$ tends to be more uniformed distributed within the boundary layer than NOx when the turbulence is strong. We

found that the NOx is uniformly distributed at noon and in the afternoon. Therefore, it is reasonable that we assume the SO2 is uniformly distributed at noon and in the afternoon.

**Comments:** Figure 4: The concentration of NOx can be more than 200 ppbv. Is it a heavy pollution day?

**Reply:** Thanks for the comment. The NOx is high because the measurement location is close to the vehicle source. We replot Fig. 4 and found that the high NOx concentration happens after 18:00 and these profiles are removed from Fig. 4. During the measurement, the NOx concentration is about 120 ppbv. It is not a heavy pollution day on January 19, 2019, and the measured mean $PM_{2.5}$ in Beijing on this day is only 47 $\mu g/m^3$.

**Comments:** Line 243-248: The vertical measurement in this study is from ground level to 240 m. However, in this section the author takes lots of words on the differences between the situations of ground level and the top of boundary layer. From Fig. 5, the [OH] didn't increase that much at 240 m compared to ground level.

**Reply:** Thanks for the comments. In this section, we assumed that the aerosols within the mixing layer were well mixed and uniformly distributed. The main purpose of our work is to propose a framework that the nucleation processing in the upper mixing layer is stronger than that at the ground. Despite that our measurement height is from ground level to 240 m, the main conclusion of the development of the mixing layer during the daytime applies to the whole mixing layer. It is reasonable we compare the differences between the situations of ground level and the top of the boundary layer.

The [OH] increased from $6.39 \times 10^6$ cm$^3$ at the ground to $11.23 \times 10^6$ cm$^3$ at the top of the mixing layer by 76.7% and increased from $6.39 \times 10^6$ cm$^3$ at the ground to $9.3 \times 10^6$ cm$^3$ at 240 m height by 44.8% when the mixing layer type is B with a mixing layer height of 500 m. Thus the [OH] increased significantly with height.

**Comments:** Line 247-254: I don't think a schematic graph is needed here. Moreover, the schematic graph is not well presenting the author's view. The loop showing in Fig. 6 is a positive feedback loop, but I think it is not the case in this study.

**Reply:** Thanks for the comments. We agree with the reviewer that Fig. 6 is not needed and we removed Fig. 6 in the manuscript.

**Comments:** Section 3.4: In this section, the author discussed the vertical profiles of photolysis rates for 4 types of aerosol profiles. But most of discussions are about the comparisons between the top of boundary layer and ground, which is not related to the measurement of this study. The author also needs to consider the vertical distribution of SO2, CS, VOCs when discussing the reasons of the NPF occurring at high altitude.

**Reply:** Thanks for the comments. In this section, we assumed that the aerosols within the mixing layer were well mixed and uniformly distributed. The main purpose of our work is to propose a framework that the nucleation processing in the upper mixing layer is stronger than that at the ground. Despite that our measurement is from ground level to 240 m, the main conclusion of the development of the mixing layer during the daytime applies to the whole mixing layer. It is reasonable we compare the differences between the situations of

ground level and the top of the boundary layer. In our framework, we focus on the influence of aerosol-radiation interaction on the nucleation processing within the mixing layer, the SO2s, CS, and VOCs are assumed to be uniformly distributed within the mixing layer.

Du, P., Gui, H., Zhang, J., Liu, J., Yu, T., Wang, J., Cheng, Y., and Shi, Z.: Number size distribution of atmospheric particles in a suburban Beijing in the summer and winter of 2015, Atmospheric Environment, 186, 32-44, 10.1016/j.atmosenv.2018.05.023, 2018.

Elterman, L., Wexler, R., and Chang, D.: COMPARISON OF AEROSOL MEASUREMENTS OVER NEW MEXICO WITH ATMOSPHERIC FEATURES, Journal of the Optical Society of America, 58, 741-&, 1968.

Qi, X., Ding, A., Nie, W., Chi, X., Huang, X., Xu, Z., Wang, T., Wang, Z., Wang, J., Sun, P., Zhang, Q., Huo, J., Wang, D., Bian, Q., Zhou, L., Zhang, Q., Ning, Z., Fei, D., Xiu, G., and Fu, Q.: Direct measurement of new particle formation based on tethered airship around the top of the planetary boundary layer in eastern China, Atmospheric Environment, 209, 92-101, 10.1016/j.atmosenv.2019.04.024, 2019.

Steinfeld, J. I.: Atmospheric Chemistry and Physics: From Air Pollution to Climate Change, Environment: Science and Policy for Sustainable Development, 40, 26-26, 10.1080/00139157.1999.10544295, 1998.

Wang, Q., Sun, Y., Xu, W., Du, W., Zhou, L., Tang, G., Chen, C., Cheng, X., Zhao, X., Ji, D., Han, T., Wang, Z., Li, J., and Wang, Z.: Vertically resolved characteristics of air pollution during two severe winter haze episodes in urban Beijing, China, Atmospheric Chemistry and Physics, 18, 2495-2509, 10.5194/acp-18-2495-2018, 2018.

Zhu, X., Tang, G., Guo, J., Hu, B., Song, T., Wang, L., Xin, J., Gao, W., Münkel, C., Schäfer,

K., Li, X., and Wang, Y.: Mixing layer height on the North China Plain and meteorological evidence of serious air pollution in southern Hebei, Atmospheric Chemistry and Physics, 18, 4897-4910, 10.5194/acp-18-4897-2018, 2018.

Response to reviewer#2

Thanks for the reviewer's helpful suggestions! The comments are addressed point-by-point and responses are listed below.

**Comments:** The research on new particle formation (NPF) is of great importance considering its vital role in modulating the cloud condensation nuclei. While NPF has been widely studied, the authors primarily focus on the development of aerosol size distribution among different altitudes as well as the impact of aerosol-radiation on NPF. The analysis angle is unique and the manuscript is well written. I have a few comments prior to the acceptance of the manuscript for publication.

**Reply:** Thanks for the comments.

**Comments:** Major comments: 1. A typical NPF figure (dN/dlogDp vs. time) at a few selected altitudes might be helpful for readers to understand the evolution of NPF along with time and altitude.

2.In the abstract, the authors hypothesized that the nucleation processing in the upper boundary layer should be stronger than that at the ground. Since the observational PNSD in the vertical direction is available, I am wondering whether the authors can calculate the formation rate directly. In this way, one can easily identify whether the nucleation processing over the upper altitude is higher than that at the ground.

**Reply:** Thanks for the comment. The reviewer gave insightful suggestions for better understanding the evolution of NPF. As shown in Fig. 2 and Fig. S3, we measured only 16 PNSD profiles on January 19, 2019. The time interval is so large that the typical NPF figure

at different altitudes is not available. Calculating the formation rate is not appropriate as the measured aerosol PNSD changed significantly (Cai and Jiang, 2017) with the development of the mixing layer.

Minor comments:

**Comments:** Line 120: Some changes were made in the source code of the TUV model so that the model can calculate the J(NO2) and J(O1D) profiles. A bit more information is useful. For instance, a few more words on what major changes have been made.

**Reply:** Thanks for the comment. We added a brief description of the changes of the TUV model in the manuscript.

In the TUV model, the input of the aerosol optical properties are the aerosol optical depths at the wavelength of 550 nm and the column-averaged SSA. The profiles of the $\sigma_{sca}$ are calculated assuming that the aerosol $\sigma_{sca}$ are proportional to those measured by Elterman et al. (1968). The g values are set to be fixed as 0.61. Some changes were made in the source code of the TUV model. In our model, the author-defined aerosol $\sigma_{sca}$ profiles, SSA profiles and g profiles can be used as the input of the model. Therefore, the $J(NO_2)$ and $J(O^1D)$ profiles with different aerosol optical profiles (including aerosol $\sigma_{sca}$, SSA, and g) can be estimated.

**Comments:** Line 123: Altitude, Please revise to altitudes

**Reply:** Thanks for the comment. We revised the word.

**Comments:** Line 132 (Line 193; 205;304): Transportation Please revise to transport.

**Reply:** Thanks for the comment. We revised the corresponding words.

**Comments:** Line 144 for different altitudes and a different time. Please revise to for different altitudes and time

**Reply:** Thanks for the comment. We revised this sentence.

**Comments:** Line 159 These particles were still not well mixed at the range between 0 and 240 m until 11:20. From Fig. 2, it does not seem to show the mixing at 11:20. Is it something not shown in the figure?

**Reply:** Thanks for the comment. The time should be 10:20. We revised the text in the corresponding manuscript.

**Comments:** Line 183 (Line 479;295) different times, Please revise to: different time

**Reply:** Thanks for the comment. We revised the corresponding words.

**Comments:** Line 185: The ratio on the ground surface decreased. It is not clear that the ground decreased compared to ??

**Reply:** Thanks for the comment. We want to say that the ratio at the ground decreased over time during 8:00 and 10:00 in the morning.

**Comments:** Line 196 later Should be late; if the authors only want to emphasize afternoon, the word "later" should be removed.

**Reply:** Thanks for the comment. We revised the words into "late afternoon".

**Comments:** Line 266-270 The authors list the altitude of boundary layer height of either 1000 or 500 meters. Please clarify whether these heights are the exact boundary layer heights or approximate altitude? Or they represent a range, i.e., less than 1000 meters?

**Reply:** Thanks for the comment. The boundary layer heights of 1000 or 500 meters are the exact boundary layer heights. We made some revisions in the corresponding lines.

Cai, R., and Jiang, J.: A new balance formula to estimate new particle formation rate: reevaluating the effect of coagulation scavenging, Atmospheric Chemistry and Physics, 17, 12659-12675, 10.5194/acp-17-12659-2017, 2017.

Elterman, L., Wexler, R., and Chang, D.: COMPARISON OF AEROSOL MEASUREMENTS OVER NEW MEXICO WITH ATMOSPHERIC FEATURES, Journal of the Optical Society of America, 58, 741-&, 1968.

**Impact of aerosol-radiation interaction on new particle formation**

[revised manuscript text omitted]

measured using a scanning mobility particle size (SMPS; TSI Inc. 3010) every five minutes. Aerosol

scattering coefficient ($\sigma_{sca}$) at the wavelength of 450 nm, 525 nm, and 635 nm were measured by an

Aurora 3000 nephelometer (Müller et al., 2011) with a time resolution of one minute. The nitrogen

dioxide ($NO_2$) was measured every minute based on its absorbance at 405 nm with a low-power

90 lightweight instrument (model 405 nm, 2B Technology, USA). The nitrogen monoxide (NO) was

measured by adding an excess of ozone with another power lightweight instrument (model 106-L,

2B Technology, USA) with a time resolution of one minute. The wind speed, wind direction,

ambient relative humidity, and temperature were measured by a small auto meteorology station. This

instrument can record the atmosphere pressure, which was used to retrieve the altitude information.

95 All of the data were averaged with a time resolution of five minutes.

**2.2 Lognormal fit of PSND**

For each of the measured PNSD, it is fitted by three lognormal distribution modes by:

$$N(Dp) = \sum_{i=1,2,3} \frac{N_i}{\sqrt{2\pi}\log(\sigma_{g,i})} exp\left[-\frac{\log(Dp)-\log(Dp_i)}{2log^2(\sigma_{g,i})}\right] \tag{31}$$

Where $N_i$, $\sigma_{g,i}$, and $Dp_i$ are the number concentration, geometric standard deviation, and geometric mean diameter of mode $i$ respectively. Two examples of fitting the measured PNSD are shown in Fig. S1. The three modes with geometric diameter ranges of 10 ~ 25 nm, 25 ~100 nm, and 100 ~ 700 nm correspond to the nucleation mode, Aitken mode, and accumulation mode respectively. The nucleation particles mainly result from the nucleation process and the Aitken mode particles are from primary sources, such as traffic sources (Shang et al., 2018). The accumulation mode particles are correlated with secondary formation, which mainly represents the ambient pollution conditions (Wu et al., 2008).

**2.3 Mie Model**

Mie scattering model (Bohren and Huffman, 2007) is used to estimate the aerosol optical properties. When running the Mie model, aerosol PNSD, aerosol black carbon mass size distribution and refractive index are essential. The measured mean black carbon mass size distribution from Zhao et al. (2019) is adopted in this study, which is measured around 3 kilometers away from this site. The refractive index of the non-black carbon and black carbon aerosol component are 1.64+0i, which is the measured mean aerosol refractive index measured at Beijing (paper in preparation), and 1.96 + 0.66i (Zhao et al., 2017) respectively. The aerosol hygroscopic growth is not considered here because

the ambient relative humidity during the measurement was all the way lower than 30% as shown in

fig. 1(b). With the measured different aerosol PNSD and above-mentioned information, we can

calculate the corresponding aerosol optical properties, which contain the aerosol $\sigma_{sca}$, aerosol single

scattering albedo (SSA) and asymmetry factor (g).

**2.4 TUV Model**

The Tropospheric Ultraviolet-Visible radiation model (TUV), developed by Madronich and

Flocke (1997), is an advanced transfer model with an eight-stream, discrete ordinate solver. This

model can calculate the spectral irradiance, spectral actinic flux, and photo-dissociation frequencies

in the wavelength range between 121 nm and 735 nm. In this study, the photolysis frequency of the

nitrogen dioxide (J(NO$_2$)) and ozone (J(O$^1$D)) were used for further study. Inputs of the TUV model

are the aerosol optical depth and single scattering albedo (Tao et al., 2014). The cloud aerosol optical

depth is set to be zero in this study. The output of the TUV model includes the profiles of J(NO$_2$) and

J(O$^1$D).

In the TUV model, the inputs of the aerosol optical properties are the aerosol optical depths at the

wavelength of 550 nm and the column-averaged SSA. The profiles of the $\sigma_{sca}$ are calculated

assuming that the aerosol $\sigma_{sca}$ are proportional to those measured by Elterman et al. (1968). The g

values are set to be fixed as 0.61. Some changes were made in the source code of the TUV model. In

our model, the author-defined aerosol $\sigma_{sca}$ profiles, SSA profiles and g profiles can be used as the

input of the model. Therefore, the J(NO$_2$) and J(O$^1$D) profiles with different aerosol optical profiles

(including aerosol $\sigma_{sca}$, SSA, and g) can be estimated.

135

**2.5 Influence of photolysis ratio on the [H₂SO₄]**

[revised manuscript text omitted]

B.